# On Concept-Based Explanations in Deep Neural Networks

## Abstract

Deep neural networks (DNNs) build high-level intelligence on low-level raw features. Understanding of this high-level intelligence can be enabled by deciphering the concepts they base their decisions on, as human-level thinking. In this paper, we study concept-based explainability for DNNs in a systematic framework. First, we define the notion of completeness, which quantifies how sufficient a particular set of concepts is in explaining a model's prediction behavior. Based on performance and variability motivations, we propose two definitions to quantify completeness. under degenerate conditions, our method is equivalent to Principal Component Analysis. Next, we propose a concept discovery method that considers two additional constraints to encourage the interpretability of the discovered concepts. We use game-theoretic notions to aggregate over sets to define an importance score for each discovered concept, which we call *ConceptSHAP*. On specifically-designed synthetic datasets and real-world text and image datasets, we validate the effectiveness of our framework in finding concepts that are complete in explaining the decision, and interpretable.

## 1 Introduction

Deep neural networks (DNNs) have shown great success in numerous tasks (Goodfellow et al., 2016), from understanding images (Zoph et al., 2017) to answering questions (Devlin et al., 2018). Yet, in many scenarios their lack of explainability serves as a bottleneck against their real-world impact, especially in high-stake decisions such as in medicine, transportation, and finance, where such explanations help identify systematic failure cases, comply with regulations, and provide feedback to model builders. This has thus led to increasing interest in human-like explanations of DNNs.

The most commonly-used methods to explain DNNs explain each prediction by quantifying the importance of each input feature (Ribeiro et al., 2016; Lundberg & Lee, 2017). However, such explanations typically explain the behavior locally for each case, rather than globally explaining how the model makes its decisions. Also, input features (such as the raw pixel values), and weights on them, are not necessarily the most effective explanations for human understanding. Instead, "concept-based explanations" characterize the global behavior of a DNN in a way understandable to humans, by explaining how DNNs use concepts in arriving at particular decisions. Such concept-based thinking, by extracting similarities from numerous examples and grouping them systematically based on their resemblance, has been shown to play an essential role in human minds for making generalizations (Armstrong et al., 1983; Tenenbaum, 1999). With a similar motivation, "concepts" can explain the decision-making rationale of DNNs and their generalizable knowledge. A few recent studies have thus focused on bringing such concept-based explainability to DNNs. Based on the common implicit assumption that the concepts should lie in certain linear subspaces of some intermediate DNN activations, they aim to find such concepts efficiently and relate them to data. These have ranged from supervised approaches (Kim et al., 2018; Zhou et al., 2018) that obtain concept representations given human-labeled data on salient concepts, to purely unsupervised approaches that provide concept explanations automatically without human labeling, ranging from k-means clustering of DNN activations (Ghorbani et al., 2019), to a self-interpretable Bayesian generative model (Bouchacourt & Denoyer, 2019). A key motivating question we ask in this paper is whether we could build on such unsupervised approaches to extract concepts, but where in addition to ensuring that the concepts are representative of the DNN activations, we would also like to ensure the additional facet that they are sufficiently predictive of the DNN function itself.

This leads naturally to a crucial unanswered question in concept-based explanation, which is how to evaluate whether a set of concepts are sufficient for prediction. Previous concept-based explanations

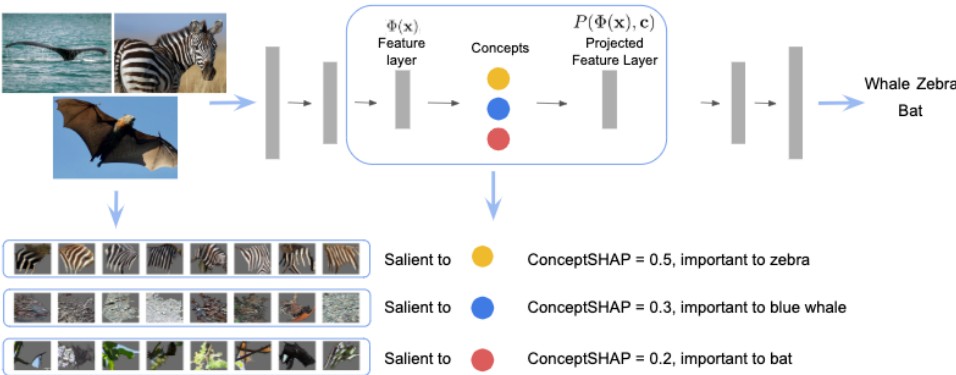

Figure 1: The overview of our concept discovering algorithm. Given a deep classification model, we first provide semantically meaningful clusters by segmentation followed by k-means clustering as in Ghorbani et al. (2019). Then, we discover complete and interpretable concepts under the constraint that each concept is salient to one (or a few) unique cluster, while projecting features onto the span of concept vectors does not deteriorate the classification performance. After the concepts of interest are retrieved, we can calculate the importance of each concept and the classes where each concept is the most important by ConceptSHAP.

select concepts that are salient to a particular class (Kim et al., 2018). However, selecting a set of salient concepts does not guarantee that these concepts are sufficient for prediction. The notion of explanations that are sufficient for prediction is also called the "completeness" of explanations (Gilpin et al., 2018), which is acknowledged to be valuable for evaluating explanations (Yang et al., 2019). In this work, we propose such a completeness metric for a given set of concept explanations. The completeness measurement can be applied to a set of concept vectors that lie in the span of some intermediate DNN layer activations, which is a general assumption in previous concept-based explanation works (Kim et al., 2018). The core idea is that, by projecting the activations onto the span of concept vectors, we keep just that information that can be explained by the concepts, and discard the information that are orthogonal to all concepts. Thus, when projecting activations onto the span of concept activation vectors result in no loss in prediction accuracy, we can learn concepts that are "complete" (i.e. sufficient for prediction).

Interestingly, we show that under a stringent degeneracy condition on the DNNs, principal component analysis (PCA) on the DNN activations can be shown to maximize these concept completeness metrics. Of course such degeneracy assumptions likely do not hold in general, so that maximizing these completeness metrics could be viewed as a generalization of PCA that additionally takes the DNN model into account. However the resulting "principal components" are not guaranteed to be interpretable to humans. We thus build on the concept-interpretability principles proposed in Ghorbani et al. (2019), and additionally consider carefully designed objectives that favors concepts that are more semantically meaningful to humans. A key facet of our approach is that it can work without any human supervision, which reduces the human labeling cost to provide explanations.

After a set of highly-complete concepts are discovered, we use game-theoretic notions to aggregate over sets to define contextualized importance of a concept, which we call ConceptSHAP. Concept-SHAP is shown to be the only scoring method that satisfies a set of axioms, which accounts for the contribution of each concept to the completeness score. We also derive a class-specific version of ConceptSHAP that decomposes the ConceptSHAP score with respect to each class in the multi-class classification setting, which can be used to find concepts that contribute the most with respect to a specific class. To verify the effectiveness of our completeness-aware concept discovery method, we create a synthetic dataset where we can obtain the ground truth concepts and test whether existing methods can retrieve them. We find that our method is able to retrieve the ground truth concepts better than all compared methods. We also demonstrate examples from real-world language and vision datasets to show that our concept discovery algorithm provides additional insights on the behavior of the model.

## 2 COMPLETENESS OF CONCEPTS

**Problem setting:** We are given a set of $n$ training examples $\mathbf{x}_1, \mathbf{x}_2, ..., \mathbf{x}_n \in \mathbb{R}^i$, corresponding labels $y_1, y_2, ..., y_n \in \mathbb{R}^o$, and a DNN $f(\mathbf{x})$ that is learned to map the labels (with dimension $o$)

from given inputs (with dimension $i$). We choose an intermediate layer of the DNN, and define the operation for generating the intermediate features from input as $\Phi(\mathbf{x}) \in \mathbb{R}^d$ and feed forwarding from the intermediate layer to logit layer as $h(\cdot)$, yielding the decomposition $f(\mathbf{x}) = h(\Phi(\mathbf{x}))$. We define the data matrix as $X \in \mathbb{R}^{i \times n}$; the corresponding feature matrix as $\Phi(X) \in \mathbb{R}^{d \times n}$, and the corresponding prediction matrix as $f(X) \in \mathbb{R}^{o \times n}$. Assume that there is a set of $m$ concepts denoted by vectors $\mathbf{c}_1, \mathbf{c}_2, ..., \mathbf{c}_m$ that represented linear directions in some activation space $\Phi(\cdot) \in \mathbb{R}^d$ given by a concept discovery algorithm. We define the concept matrix as $\mathbf{c} = [\mathbf{c}_1 \ \mathbf{c}_2 \ldots \ \mathbf{c}_m]$.

Next, we propose two mathematical definitions that capture the completeness of a given set of given concepts. Both definitions are based on the idea that completeness should quantify how sufficient a particular set of concepts are in explaining the model's behavior. A low completeness score of a set of concepts indicates that the corresponding concepts do not capture the model behavior fully, and that the model bases its decision on factors other than the given concepts. We propose two metrics of completenss based on two different assumptions, as we discuss below.

**Assumption 1:** If the given set of concepts is complete, then using a projection of the intermediate features from input onto the feature subspace spanned by the concepts, *concept space*, would not deteriorate the model performance. We define the projection of some input embedding $\Phi(\mathbf{x})$ onto the subspace spanned by $\mathbf{v} \in \mathbb{R}^{d \times r}$ as

$$P(\Phi(\mathbf{x}), \mathbf{v}) = \mathbf{v}(\mathbf{v}^\top \mathbf{v})^{-1} \mathbf{v}^\top \Phi(\mathbf{x}). \tag{1}$$

We define the completeness metric $\eta^{(1)}$ on a set of validation data with T data points as $V = \{(\mathbf{x}_1, y_1), ..., (\mathbf{x}_T, y_T)\}$ based on the assumption that projecting input features onto the span of a complete set of concepts should not reduce the model prediction performance.

**Definition 2.1.** Given a prediction model $f(\mathbf{x}) = h(\Phi(\mathbf{x}))$, a set of concept vectors $\mathbf{c}_1, ..., \mathbf{c}_m$, and some loss metric $L$, we define the completeness score $\eta^{(1)}$ as:

$$\eta^{(1)}(\mathbf{c}_1, ..., \mathbf{c}_m) = \frac{R - \sum_{\{\mathbf{x}, y\} \in V} L(h(P(\Phi(\mathbf{x}), \mathbf{c})), y)}{R - \sum_{\{\mathbf{x}, y\} \in V} L(f(\mathbf{x}), y)}, \tag{2}$$

where $R = \sum_{\{\mathbf{x}, y\} \in V} L(h(\mathbf{0}), y)$ to ensure that $\eta^{(1)}(\mathbf{0}) = 0$. We omit the dependency of $h(\cdot)$, $\Phi(\cdot)$, $f(\cdot)$, and $L(\cdot)$ of $\eta(\cdot)$ for notation simplicity. When $\eta^{(1)}(\mathbf{c}_1, ..., \mathbf{c}_m)$ is high, the network maintains a high accuracy even after projection, which supports that the set of discovered concepts hold sufficient information for prediction.

**Assumption 2:** The second assumption is that if we remove all useful concept information for a classification task, the model should fail to discriminate different classes. Thus, when all salient information is removed from the network, predictions scores for examples in class A won't be much different from other examples in class A. We define the data matrix of validation set as as $X_v = [\mathbf{x}_1 \ \mathbf{x}_2 \ldots \ \mathbf{x}_T]$. To quantify how much the prediction score varies across data samples, we use the sample variance of the predictions: $\hat{\text{var}}(f(X_v)) = \text{Tr}(\hat{\text{cov}}(f(X_v))) = \text{Tr}((f(X_v) - \hat{\mathbb{E}}[f(X_v)])(f(X_v) - \hat{\mathbb{E}}[f(X_v)])^\top)$, where $\hat{\mathbb{E}}[f(X_v)] = \frac{1}{T} \sum_{i=1}^{T} f(\mathbf{x}_i)$, and Tr stands for the trace. Then, we define the second completeness metric following this assumption.

**Definition 2.2.** Given a prediction model $f(\mathbf{x}) = h(\Phi(\mathbf{x}))$, and a set of concept vectors $\mathbf{c}_1, \mathbf{c}_2, ..., \mathbf{c}_m$, we define the completeness score $\eta^{(2)}$ as:

$$\eta^{(2)}(\mathbf{c}_1, ..., \mathbf{c}_m) = 1 - \frac{\hat{\text{var}}(h(\Phi(X_v) - P(\Phi(X_v), \mathbf{c})))}{\hat{\text{var}}(h(\Phi(X_v)))}, \tag{3}$$

Based on our assumption 2, the variance of the prediction gets lower after useful concept information is removed from the data, yielding a high completeness score $\eta^{(2)}$.

We now show that under degenerate assumptions, the top $k$ PCA vectors of $\Phi(\mathbf{x})$ maximize the completeness score for a set of concept vectors. Top PCA vectors are designed to capture as much information in data as possible, a set of concepts with high completeness score similarly preserve the necessary information in the data for the model to reach satisfactory predictions.

**Proposition 2.1.** *When h is an isometry function that maps from $(\Phi(\cdot), \|\cdot\|_F) \to (f(\cdot), \sqrt{L})$, where L is the loss metric in equation 2 and $f(\mathbf{x}_i) = y_i$, $\forall (x_i, y_i) \in V$ (i.e. the loss is minimized), the first m PCA vectors maximizes $\eta^{(1)}$.*

**Proposition 2.2.** *When h is an isometry function that maps from $(\Phi(\cdot), \|\cdot\|_F) \to (f(\cdot), \|\cdot\|_F)$, and each dimension of $\Phi(\mathbf{x})$ is uncorrelated with unit variance, the first m PCA vectors maximize $\eta^{(2)}$.*

We underline the two main differences between the concept vectors that maximize the completeness score and the PCA vectors. First, the propositions depend on degeneracy assumptions such as isometry of a DNN, which may not hold in practice. Therefore, the concepts that maximize the completeness score takes the prediction of the DNN into account, which can be seen as a generalization of the original PCA. Second, since the concept score only depends on the span of the set of concept vectors, any concept vectors whose span is equal to the span of the top PCA vectors also maximize the completeness score (i.e. the set of vectors that maximize the completeness is not unique). Each PCA vectors are constrained to minimize the reconstruction error and being orthogonal to other PCA directions. On the other hand, the discovered concept vectors that maximize the completeness can be designed so that each concept is interpretable and semantically-meaningful to humans, which will be further explained in the next section.

## 3 DISCOVERING COMPLETE AND INTERPRETABLE CONCEPTS

Our goal is to discover a set of maximally-complete concepts, where each concept is also interpretable and semantically-meaningful. Ghorbani et al. (2019) has listed meaningfulness, coherency, and saliency as the desired properties for concept-based explanations. Our work on completeness is a crucial addition to the set: not only concept are meaningful coherent and salient, we ensure they are sufficient to models prediction.

We assume that we are given some candidate clusters of concepts (which can be given by human labeling or self-discovery) and each cluster shares some feature attributes that are coherent and semantically-meaningful to humans (which matches the two desired properties in Ghorbani et al. (2019)). We define the feature matrix of cluster i as $\tau_i = [\Phi(\mathbf{x}_{i1})\ \Phi(\mathbf{x}_{i2})\dots]$, where $\mathbf{x}_{i1}, \mathbf{x}_{i2}$ are samples that belong to cluster i. We denote the feature mean of cluster $i$ as $\mu_i = \text{mean}(\tau_i)$. Clusters can be obtained by human labeling (Kim et al., 2018) or by unsupervised grouping of relevant input features (e.g. segmentation of images based on grouping of pixels) (Ghorbani et al., 2019). In either case, we would not know which sets of clusters contain useful information to the model that we try to explain. We aim to find a minimum set of concepts that are maximally-complete to the prediction model. Additionally, we constraint that each concept is salient to one cluster only so that each concept direction is semantically-meaningful to humans. To discriminate different concepts (for coherency), we constraint that different concepts are not salient to the same cluster.

We now define our objective function for discovering a set of complete and interpretable concepts $\mathbf{c}$. A primary goal is maximizing completeness $\eta$ (which can be $\eta^{(1)}$ or $\eta^{(2)}$), such that the set of concepts fully explain the model behavior. Besides, we introduce two regularization terms for interpretability (can be considered as generalization of the orthogonality constraint of PCA). We introduce cluster-sparsity regularization $L_{\text{sparse,Cl}}(\mathbf{c})$ to encourage each concept is salient to minimum number of clusters, and we introduce concept-sparsity regularization $L_{\text{Sparse,Con}}(\mathbf{c})$ to encourage different concepts are not salient to the same cluster, i.e. each cluster to be salient to at most one concept. Given some clusters $\tau_1, \tau_2, ..., \tau_K$, a set of training examples $\mathbf{x}_1, \mathbf{x}_2, ..., \mathbf{x}_n$, and a pre-trained prediction model $f(\mathbf{x}) = h(\Phi(\mathbf{x}))$, the overall objective function (to minimize) becomes:

$$-\eta(\mathbf{c}) + \lambda_1 \cdot L_{\text{Sparse,Cl}}(\mathbf{c}) + \lambda_2 \cdot L_{\text{Sparse,Con}}(\mathbf{c}), \tag{4}$$

where $\lambda_1$ and $\lambda_2$ are loss coefficients. To formulate the cluster-sparsity regularization $L_{\text{sparse,Cl}}(\mathbf{c})$ and concept-sparsity regularization $L_{\text{Sparse,Con}}(\mathbf{c})$, we first formally introduce the *saliency score* between concept $\mathbf{c}_j$ to cluster $\tau_k$ as:

$$\rho(\mathbf{c}_j, \tau_k) = \frac{|\langle \mathbf{c}_j, \mu_k \rangle|}{\sqrt{\sum_{l=1}^K |\langle \mathbf{c}_j, \mu_l \rangle|^2}}.$$

We note that the saliency score is normalized such that the saliency score between any concept and all clusters has unit norm. When the saliency score between concept $\mathbf{c}_j$ to cluster $\tau_k$ is large, $\mathbf{c}_j$ can differentiate samples from cluster $k$ from samples in a random cluster, and thus $\mathbf{c}_j$ is salient to $\tau_k$. To encourage that each concept can differentiate a small amount of clusters to random clusters, we regularize the L1 norm of saliency score for every concept-cluster pair (which can be seen as the

sparse filtering objective in Ngiam et al. (2011)), leading to the cluster-sparsity regularization loss:

$$L_{\text{sparse,Cl}}(\mathbf{c}) = \sum_{j=1}^{m} \sum_{k=1}^{K} \rho(\mathbf{c}_j, \tau_k),$$

which encourages sparse saliency scores. To constrain that different concepts are not salient to the same cluster, we penalize the pairwise saliency score product between every pair of concepts for the same cluster, leading to the concept-sparsity regularization loss:

$$L_{\text{Sparse,Con}}(\mathbf{c}) = \sum_{i \neq j} \sum_{k=1}^{K} \rho(\mathbf{c}_i, \tau_k) \cdot \rho(\mathbf{c}_j, \tau_k).$$

If there are two concepts that are both salient with respect to the same cluster, the pairwise saliency score will be large and thus the concept-sparsity regularization loss will be large. We note that each concept has to be salient to some cluster, but a cluster can be not salient to any concepts. Therefore, we typically assume we have more clusters compared to concepts (i.e. $K > m$).

## 4 HOW IMPORTANT IS EACH CONCEPT?

**ConceptSHAP to quantify concept importance:** Given a set of concepts $C_S = \{\mathbf{c}_1, \mathbf{c}_2, ...\mathbf{c}_m\}$ with a high completeness score, we would like to evaluate the importance of each individual concept, specifically, by quantifying how much each individual concept contributes to the final completeness score. Let $\phi_i$ denote the importance score for concept $C_i$, such that $\phi_i$ quantifies how much of the completeness score $\eta(C_S)$ is contributed by $\mathbf{c}_i$. Motivated by its successful applications in quantifying attributes in what-if scenarios for complex systems, we adapt Shapley values (Shapley, 1988; Lundberg & Lee, 2017), to fairly assign the importance of each concept (which we abbreviate as ConceptSHAP):

**Definition 4.1.** Given a set of concepts $C_S = \{\mathbf{c}_1, \mathbf{c}_2, ...\mathbf{c}_m\}$ and some completeness metric $\eta$, we define the ConceptSHAP $\phi_i$ for concept $\mathbf{c}_i$ as

$$\phi_i(\eta) = \sum_{S \subseteq C_s \backslash \mathbf{c}_i} \frac{(m - |S| - 1)!|S|!}{m!} [\eta(S \cup \{\mathbf{c}_i\}) - \eta(S)],$$

The main benefit of using Shapley value to assign importance is that Shapley value can be shown to uniquely satisfy a set of desired axioms, listed in the following proposition:

**Proposition 4.1.** *Given a set of concepts* $C_S = \{\mathbf{c}_1, \mathbf{c}_2, ...\mathbf{c}_m\}$ *and a completeness metric* $\eta$*, and some importance score* $\phi_i$ *for each concept* $\mathbf{c}_i$ *that depends on the completeness metric* $\eta$*.* $\phi_i$ *defined by conceptSHAP is the unique importance assignment that satisfy the following four axioms:*

- *Efficiency: The sum of all importance value should sum up to the total completeness value,* $\sum_{i=1}^{m} \phi_i(\eta) = \eta(C_S)$.

- *Symmetry: For two equivalent concepts, which satisfy* $\eta(u \cup \{\mathbf{c}_i\}) = \eta(u \cup \{\mathbf{c}_j\})$ *for every subset* $u \subseteq C_S \backslash \{\mathbf{c}_i, \mathbf{c}_j\}$*,* $\phi_i(\eta) = \phi_j(\eta)$.

- *Dummy: If* $\eta(u \cup \{\mathbf{c}_i\}) = \eta(u)$ *for every subset* $u \subseteq C_S \backslash \{\mathbf{c}_i\}$*, then* $\phi_i(\eta) = 0$.

- *Additivity: If* $\eta$ *and* $\eta'$ *have importance value* $\phi(\eta)$ *and* $\phi(\eta')$ *respectively, then the importance value of the sum of two completeness metric should be equal to the sum of the two importance values, i.e,* $\phi_i(\eta + \eta') = \phi_i(\eta) + \phi_i(\eta')$ *for all i.*

The efficiency axiom distributes the completeness score of all concepts to the individual concepts. The symmetry axiom guarantees that two concepts that behaves the same get the same importance score for fairness. The dummy axiom guarantees that concepts that do not affect the completeness gets 0 importance score. The additivity axiom guarantees that decomposibility in the completeness leads to decomposibility in the importance score, and scaling the completeness does not change relative importance ratio between concepts.

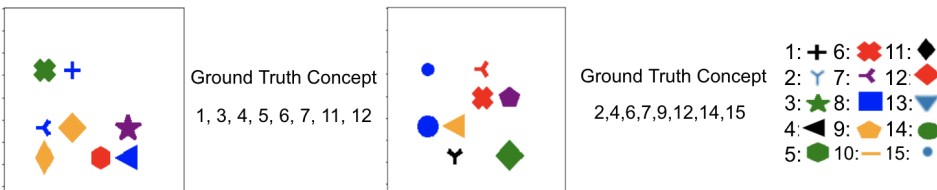

Figure 2: Two random training images and the respecting ground truth concepts that are positive along with a table that matches ground truth concepts to shape. Each object shape in the image corresponds to a ground truth concept (with random color and location), and the ground truth label depends solely on ground truth concept 1 to 5. Only the training image and ground truth label are provided during training (in the unsupervised case), and the goal of the discovering concept algorithm is to correctly retrieve ground truth concepts $\xi_1$ to $\xi_5$.

**Per-class saliency of concepts:** In multi-class classification, it may be more informative to obtain a set of related concepts that contribute to the prediction for a specific class, instead of the global contribution (i.e. concepts that are important to all classes). To obtain the concept importance score for each class, we first define the completeness score with respect to one class by only considering data points that belongs to that class, which is formalized as:

**Definition 4.2.** Given a prediction model $f(\mathbf{x}) = h(\Phi(\mathbf{x}))$, a set of concept vectors $\mathbf{c}_1, \mathbf{c}_2, ..., \mathbf{c}_m$ that lie in the feature subspace in $\Phi(\cdot)$. We then define the completeness score for class $j$ as:

$$\eta_j^{(1)}(\mathbf{c}_1, ..., \mathbf{c}_m) = \frac{R_j - \sum_{\{\mathbf{x},y\} \in V_j} L(h(P(\Phi(\mathbf{x}), \mathbf{c})), y)}{R - \sum_{\{\mathbf{x},y\} \in V} L(f(\mathbf{x}), y)}, \tag{5}$$

where $V_j$ is the set of validation data where ground truth label is $j$ and $R_j = \sum_{\{\mathbf{x},y\} \in V_j} L(h(\mathbf{0}), y)$. Given the completeness for a specific class, we define the ConceptSHAP for concept i with respect to class j as:

**Definition 4.3.** Given a prediction model $f(\mathbf{x})$, a set of concept vectors $\mathbf{c}_1, \mathbf{c}_2, ..., \mathbf{c}_m$ that lie in the feature subspace in $\Phi(\cdot)$. We can define the ConceptSHAP for concept i with respect to class j as:

$$\phi_{i,j}(\eta) = \phi_i(\eta_j). \tag{6}$$

For each class j, we may select the concepts with the highest conceptSHAP score with respect to class j. We note that $\sum_j \eta_j = \eta$ and thus with the additivity axiom, $\sum_j \phi_{i,j}(\eta_j) = \phi_i(\eta)$.

## 5 EXPERIMENTS

### 5.1 SYNTHETIC DATA WITH GROUND TRUTH CONCEPTS

**Setting:** We construct a synthetic image dataset with known complete concepts to evaluate whether the proposed automatic concept discovery algorithm can successfully extract the ground truth concept accurately. For each sample, we randomly sample 15-dimensional binary variable assigned as ground truth candidate concepts $\xi_1, ..., \xi_{15}$ that is generated with Bernoulli independently for each dimension with $p = 0.5$. From ground truth concepts $(\xi)$, we generate input data x and output label y. For the label target $y$, we construct a 15-dimensional multi-label target for each sample, where the target $y$ is a function that depends on the first 5 dimension of the 15-dimensional $\xi$. For example, $y_1 = \sim (\xi_1 \cdot \xi_3) + \xi_4, y_2 = \xi_2 + \xi_3 + \xi_4, y_3 = \xi_2 \cdot \xi_3 + \xi_4 \cdot \xi_5$[1]. Therefore, the minimum set of ground truth variable is $\{\xi_1, ..., \xi_5\}$ by construction. For the input data $\mathbf{x}$, we construct a toy image dataset where each concept $\xi_i$ is mapped to a specific shape, and the image contains the specific shape if and only if the concept $\xi_i = 1$. For example, if $\xi_{3t} = 1$, a star (with random color and location) will occur in the image $\mathbf{x}_t$, and if $\xi_{3t} = 0$, there will be no star in the image $\mathbf{x}_t$. The map of concept to shape and two example images are given in Figure 2.

For the input cluster image for our discover concept algorithm, we either provide the ground truth clustering or by superpixel segmentation followed by K-means clustering as in Ghorbani et al. (2019), which we call the method as ours-supervised and ours-unsupervised respectively. In total, we use 48k training samples and 12k evaluation samples, where each ground truth concept corresponds to some specific shape in the image. We train a convolutional neural network with 6 layers which achieves 0.999 accuracy, and take the first fully connected layer as the feature layer (which is $\Phi(\mathbf{x})$ in the problem definition.)

[1] $\sim$ notes Not, and the details of generating this dataset is in the appendix.

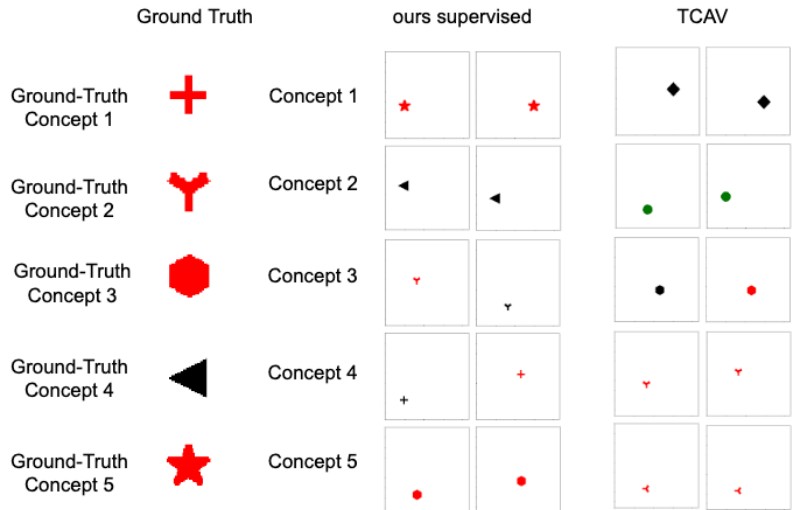

Figure 3: Visualization Result for the nearest neighbors of each discovered concepts in ours-supervised and TCAV along with ground truth concept 1 to 5 that is constructed to be the minimum set of ground truth variable. We note that only the shape is revelent of the concept, as the color and location can be random. We show that each of our discovered concepts in ours-supervised corresponds to one of ground truth concept 1 to 5 (with a random order). While TCAV also shows meaningful discovered concepts, they fail to retrieve all ground truth concepts that are used by the model. Higher resolution examples will be shown in the appendix due to space constraint.

**Evaluation metrics:** Let the known concepts be $\xi_1$, $\xi_2$, ..., $\xi_{\hat{m}}$, and assume we discover some concept vectors $\mathbf{c}_1$, ..., $\mathbf{c}_m$. We would like to evaluate how closely the discovered concept vectors align with the actual ground truth concepts. For a concept vector $\mathbf{c}_i$ to align with a ground truth concept $\xi_j$, we assume that the ground truth concept can be linearly separated by the concept vector direction. More formally, we measure the accuracy of the best linear classifier with $\mathbf{c}_i$ as the weight vector applied on the binary classification problem where $\xi_j$ is the target.

$$\text{Score}(\mathbf{c}_i, \xi_j) = \max_{a \in \{-1,1\}, b \in \mathbb{R}} \sum_{t=1}^{T} \frac{\mathbb{1}\left[(a \cdot \mathbf{c}_i^T \Phi(\mathbf{x}_t) > b) == \xi_{jt}\right]}{T}.$$

We then evaluate how well the set of discovered concepts $\mathbf{c}_1...\mathbf{c}_m$ matches the set of ground truth concepts $\xi_1$, ..., $\xi_{\hat{m}}$ as

$$\text{AlignemntScore}(\cup_{i=1}^{m}\{\mathbf{c}_i^{'}\}, \cup_{j=1}^{\hat{m}}\{\xi_j\}) = \frac{1}{\hat{m}+1} \max_{P \subseteq [1,m]^m} \sum_{j=1}^{\hat{m}} \text{Acc}(\mathbf{c}_{P[j]}, \xi_j),$$

which measures the best average accuracy by assigning the best concept vector to differentiate each ground truth concept.

**Results:** We summarize the results in Table 1, where ours-supervised and TCAV takes supervised clusters as input, and ours-unsupervised, ACE, Raw-Clustering takes the clustered segments as input. For supervised clusters, we randomly choose examples where $\xi_j = 1$ for cluster j. The term supervised and unsupervised refers to whether the actual ground truth concept set $\xi_j$ is given or not. For ours-supervised 1, we maximize $\eta^{(1)}$ in equation 4; for ours-supervised 2, we maximize $\eta^{(2)}$ in equation 4. We see that both ours-supervised 1 and ours-supervised 2 obtain higher AlignemntScore compared to TCAV. ours-unsupervised 1 and ours-unsupervised 2 also achieves higher AlignemntScore than all compared baselines, which demonstrates the effectiveness of our concept discovery algorithm. We further observe that that completeness 1 and 2 are complementary: maximizing completeness 1 does not necessary lead to a higher value in completeness 2, and vice versa. Nevertheless, by jointly optimizing completeness 1 or completeness 2 along with additional sparsity regularization with respect to given clusters, we are able to retrieve the correct ground truth concepts. Lastly, we show the nearest neighbors (of the super-pixel segments) for the discovered concepts of ours supervised and TCAV along with the ground truth concepts in Figure 3 to validate that our concept discovering algorithm does retrieve the correct concept. While we only show the top-2 nearest neighbors, we note that the top-k nearest neighbors examples all belong to the same concept when k is large.

| Methods | $\eta^{(1)}$ | $\eta^{(2)}$ | AlignemntScore |
|---|---|---|---|
| ours-supervised 1 | 0.96 | 0.20 | 0.90 |
| ours-supervised 2 | 0.0 | 1.0 | 0.94 |
| TCAV | 0.0 | 0.30 | 0.82 |
| ours-unsupervised 1 | 0.97 | 0.22 | 0.89 |
| ours-unsupervised 2 | 0.12 | 0.99 | 0.90 |
| ACE | 0.0 | 0.37 | 0.75 |
| PCA | 0.0 | 0.67 | 0.79 |
| Raw Clustering | 0.49 | 0.83 | 0.66 |

Table 1: The Completeness and AlignemntScore for our methods compared to the baseline methods on synthetic dataset where ground truth can be obtained.

## 5.2 TEXT CLASSIFICATION

**Setting:** We apply our method on the IMDB text classification dataset. The IMDB dataset contains text of 50k movie reviews, where 25k reviews is used as training data and 25k reviews are used for evaluation. For each review, it is either classified as a positive or negative review. We use a pre-trained model with a BERT language model (Devlin et al., 2018) from Keras, which achieves 0.94 testing accuracy. To obtain the input cluster, we use a 10-word sliding window to obtain sub-sentences over the IMDB sentences. We then obtain the embedding for all sub-sentences, and perform k-means clustering on the positive sub-sentences and negative sub-sentences. We then run our concept discovering algorithm to obtain 5 concepts with $\eta^{(1)}$ 0.99.

| Concepts | Nearest Neighbors | ConceptSHAP | Related Class |
|---|---|---|---|
| Concept 1 | plot is boring the characters are neurotic needlessly offensive characters jess bhamra parminder nagra and jules paxton keira average chop socky all of the cast are likeable characters | 0.13 | neg |
| Concept 2 | that keeps on reappearing to the scene where you think she deserved a more studied finale than that i think think no sometimes hatred and isolation are deeper are more | 0.29 | neg |
| Concept 3 | i think the most frustrating thing is that the performances you might think to see organs yanked out of the many people think has an underlying meaning the love between | 0.15 | neg |
| Concept 4 | don't wait for it to be a classic watch it has real potential and will be one to watch in i recommend you to watch it if you like mature | 0.43 | pos |
| Concept 5 | children trying to comfort them after that is all said paid so well after all acting is one of the it after watching it you will say that it was | 0.21 | pos |

Table 2: Concepts and their nearest neighbors, ConceptSHAP values, and related class in IMDB.

**Results:** For the 5 discovered concepts, we show the top nearest neighbors to each concept, and the ConceptSHAP value and related class (determined by TCAV score) for each concept discovered. Additional nearest-neighbor examples are shown in the appendix. We note that for all concepts, the nearest sub-sentences of other concepts mostly contain a specific word, which we highlight in blue. Nearest neighbors of concept 1 mostly contains the word "characters", nearest neighbors of concept 2 and concept 3 mostly contains the word "think", nearest neighbors of concept 4 mostly contains the word "watch", and nearest neighbors of concept 5 mostly contains the word "after". With a closer look at each concept's nearest neighbors, we find that the nearest sub-sentences of the first concept usually contains negative adjectives alongside "characters", nearest sub-sentences of the second concept usually contains the word "think" at the first or last position followed by disagreement towards the movie, nearest sub-sentences of the third concept usually contains "think" in the middle of the sub-sentence followed by the reviewer's more neutral personal opinion, the nearest sub-sentences of the fourth concept often contain the phrase "watch it" where "it" refers to the movie, and the nearest sub-sentences of the fifth concept just contains the word "after". We find that the most salient concept by ConceptSHAP value is the concept 4, where all of the top nearest

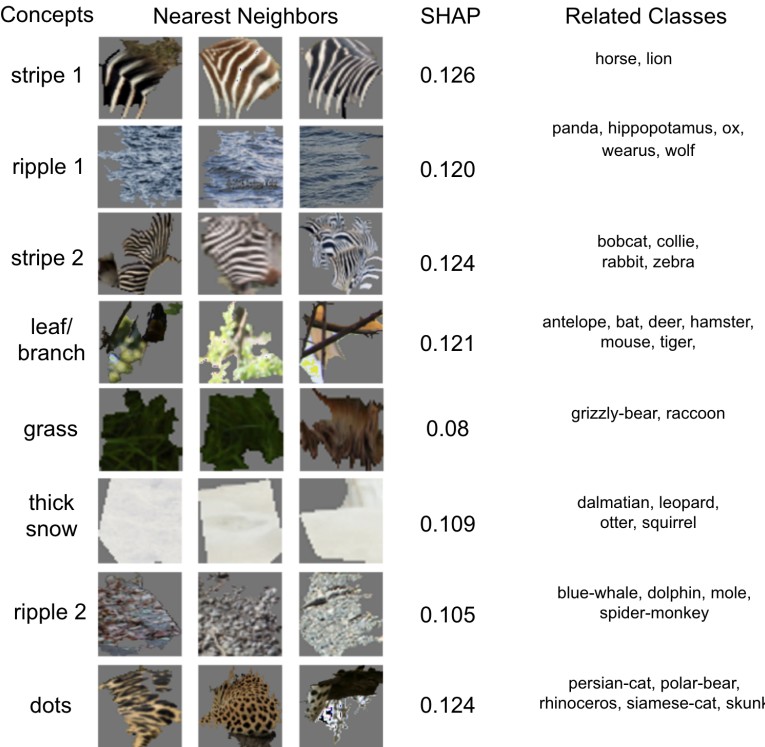

| Concepts | Nearest Neighbors | SHAP | Related Classes |
|---|---|---|---|
| stripe 1 | | 0.126 | horse, lion |
| ripple 1 | | 0.120 | panda, hippopotamus, ox, wearus, wolf |
| stripe 2 | | 0.124 | bobcat, collie, rabbit, zebra |
| leaf/ branch | | 0.121 | antelope, bat, deer, hamster, mouse, tiger, |
| grass | | 0.08 | grizzly-bear, raccoon |
| thick snow | | 0.109 | dalmatian, leopard, otter, squirrel |
| ripple 2 | | 0.105 | blue-whale, dolphin, mole, spider-monkey |
| dots | | 0.124 | persian-cat, polar-bear, rhinoceros, siamese-cat, skunk |

Figure 4: The Nearest Neighbors, ConceptSHAP, and related class for each concept obtained in AwA.

neighbors explicitly mentioned the word "watch" with a positive sentiment in general. We perform TCAV test for all concepts with respect to the positive and negative class, and the first 3 concepts are significant to the class "negative" with TCAV score 1, and the last 2 concepts are significant to the class "positive" with TCAV score 1.

## 5.3 IMAGE CLASSIFICATION

**Setting:** We next perform experiments on Animals with Attribute (AwA) (Lampert et al., 2009) to classify animals with 50 classes, where we take 26905 images as training data and 2965 images as evaluation data. Each training data has a ground truth label of one of 50 animals. We train an Inception-V3 model pre-trained on Imagenet (Szegedy et al., 2016) which reaches $0.94$ testing accuracy. To obtain the input clusters, we employ the method of Ghorbani et al. (2019), which performs superpixel segmentation and k-means clustering with images to get 334 input clusters. We then perform our discovering concepts algorithm given the clusters to obtain 8 concepts with $\eta^{(1)}$ 0.99.

**Results:** For each of the 8 discovered concepts, we show the top nearest neighbor patches, the ConceptSHAP value, and the related classes where the concept has at least twice as large ConceptSHAP value than any other concepts. From the nearest neighbor of each concept, we find that the concepts learned by the network mostly consider textures and colors. Since we only learn 8 concepts for 50 classes, each concepts learned are useful to multiple classes. We find that the ripple texture that is the most common in ocean is significant to many marine animals. The leaf/ grass concepts are often significant to animals that live in trees or pastures. We note that out of the 8 concepts learned, there are two concepts representing stripes and two concepts representing ripples. While the concept "stripe 1" seems to contain thicker stripes compared to "stripe 2", we do not observe significant difference between the top nearest neighbors of "ripple 1" and "ripple 2". Other than this, each discovered concept seems to be meaningful and coherent to humans. We note that in some cases the related class of a concept may not necessarily contains the concept. One possible reason is that the concepts may be salient since they are "pertinent negative" to a certain class, which helps making the correct prediction since these concepts do not exist in images of a certain class. The main takeaway of this example is that the salient concepts for image classification shares similarity in texture instead of shape, which coincides with the finding in Geirhos et al. (2018).

## 6 RELATED WORK

Various approaches have been proposed to explain the decision making of pre-trained models. Most works fall under two categories: (i) feature-based explanation methods, that attribute the decision to important input features (Ribeiro et al., 2016; Lundberg & Lee, 2017; Smilkov et al., 2017; Chen et al., 2018), and (ii) sample-based explanation methods, that attribute the decision to previously observed samples (Koh & Liang, 2017; Yeh et al., 2018; Khanna et al., 2019; Arik & Pfister, 2019). Among these forms of interpretability, different evaluations of explanations are proposed, including more human-centric evaluations (Lundberg & Lee, 2017; Kim et al., 2018) and functional ly-grounded evaluations (Samek et al., 2016; Kim et al., 2016; Ancona et al., 2017; Yeh et al., 2019). However, providing the most important input features or samples for a specific prediction does not necessary give insights on how the model behaves globally, which our work aims to address with concept-based explanations. For concept-based explanations, few recent works are related. TCAV (Kim et al., 2018) use human-labeled data and estimates the importance of a concept with respect to a specific class. Zhou et al. (2018) decompose the prediction of a data sample into linear combinations of concept components. Ghorbani et al. (2019) automate TCAV by replacing human-labeled data by automatically super-pixel segmentation followed by k-means clustering. Bouchacourt & Denoyer (2019) discover concept by training a inherently explainable model which trains a concept classifier along with the prediction model. While all aforementioned works defines concept directions in the linear span of some activation layer of the model, our framework brings completeness and interpretability to concept discovery.

Our work is also closely related to methods that perform dimension reduction in neural network layers to obtain meaningful latent variables and understand neural network. Chan et al. (2015) cascade PCA layers to obtain satisfactory prediction performances. Raghu et al. (2017) apply SVD followed by CCA to compare two representations of a deep model to help better understand the deep representations. Kingma & Welling (2013) perform deep dimension reduction for generative models where the latent space can be semantically-meaningful. For example, Chorowski et al. (2019) show that when learning with speech data, the latent dimension is closely related to the phonemes, which can be seen as human-relatable concepts in speech data; or Radford et al. (2017) show that when learning with language data, a single unit is closely related to the sentiment.

## 7 CONCLUSIONS

Concept-based explanations can be a key direction to understand how DNNs make decisions. In this paper, we study concept-based explainability in a systematic framework. First, we define the notion of completeness, which quantifies how sufficient a particular set of concepts is in explaining the model's behavior. Based on performance and variability motivations, we propose two definitions to quantify completeness. We show that they yield the commonly-used PCA method under certain assumptions. Next, we study two additional constraints to ensure the interpretability of discovered concept. Through experiments in toy data, text, and image domain, we demonstrate that our method is effective in finding concepts that are complete (in explaining the model's prediction) and that are interpretable. Note that although our work focuses on post-hoc explainability of pre-trained DNNs, joint training with our proposed objective function can also be used to train an inherently-interpretable model. A future direction may be to explore whether jointly learning the concepts and the model can lead to better interpretability.

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

## APPENDIX A   PROOF

**Proof of Proposition 2.1**

*Proof.* By the basic properties of PCA, the first $m$ PCA vectors (principal components) minimize the reconstruction $\ell_2$ error. Define the concatenation of the $m$ PCA vectors as a matrix $\mathbf{p}$ and $\|\cdot\|$ as the $\ell_2$ norm, the basic properties of PCA is equivalent to that for all $\mathbf{c} = [\mathbf{c}_1 \, \mathbf{c}_2 \ldots \, \mathbf{c}_m]$,

$$\sum_{\mathbf{x}\subseteq V_X} \|P(\Phi(\mathbf{x}),\mathbf{p}) - \Phi(\mathbf{x})\|_F^2 \leqslant \sum_{\mathbf{x}\subseteq V_X} \|P(\Phi(\mathbf{x}),\mathbf{c}) - \Phi(\mathbf{x})\|_F^2.$$

By the isometry of $h$ from the Frobenius norm to $\sqrt{L}$, we have

$$\sum_{\mathbf{x}\subseteq V_X} L(h(P(\Phi(\mathbf{x}),\mathbf{p})), h(\Phi(\mathbf{x}))) \leqslant \sum_{\mathbf{x}\subseteq V_X} L(h(P(\Phi(\mathbf{x}),\mathbf{c})), h(\Phi(\mathbf{x}))),$$

and since $f(\mathbf{x})$ is equal to Y, we can rewrite to

$$\sum_{\mathbf{x},y\subseteq V} L(h(P(\Phi(\mathbf{x}),\mathbf{p})), y) \leqslant \sum_{\mathbf{x},y\subseteq V} L(h(P(\Phi(\mathbf{x}),\mathbf{c})), y)$$

and subsequently get that for any $\mathbf{c}$

$$\frac{R - \sum_{\{\mathbf{x},y\}\subseteq V} L(h(P(\Phi(\mathbf{x}),\mathbf{p})), y)}{R - \sum_{\{\mathbf{x},y\}\subseteq V} L(f(\mathbf{x}), y)} \geqslant \frac{R - \sum_{\{\mathbf{x},y\}\subseteq V} L(h(P(\Phi(\mathbf{x}),\mathbf{c})), y)}{R - \sum_{\{\mathbf{x},y\}\subseteq V} L(f(\mathbf{x}), y)}.$$

$\square$

**Proof of Proposistion 2.2**

*Proof.* We note that the completeness only depends on the span of $\mathbf{c}_1, \ldots \mathbf{c}_m$. If we assume the matrix $\mathbf{c}$ to have rank $m' \leqslant m$, we may find a set of orthonormal basis (by QR decomposition) $\mathbf{c}_1, \ldots \mathbf{c}_{m'}$ that is orthonormal with the same completeness score. Therefore, for any set of given concepts $\mathbf{c}_1, \ldots \mathbf{c}_m$, we can replace them with a set of orthonormal concepts $\mathbf{c}_1, \ldots \mathbf{c}_{m'}$ without loss of generality. By the basic properties of PCA, the first m PCA vectors $\mathbf{p}_1, \ldots, \mathbf{p}_m$ maximizes the total projection data variance on the projected space with at most m orthonormal vectors, which can be formalized as

$$\sum_{i=1}^{m} \hat{\mathrm{var}}(\Phi(X_v)^\top \mathbf{p}_i) \geqslant \sum_{i=1}^{m'} \hat{\mathrm{var}}(\Phi(X_v)^\top \mathbf{c}_i), \tag{7}$$

By using the notation $\mathbf{c}^{(j)}$ for the $j^{th}$ entry of vector $\mathbf{c}$, we may rewrite total projected variance as

$$
\begin{aligned}
\sum_{i=1}^{m'} \hat{\mathrm{var}}(\Phi(X_v)^\top \mathbf{c}_i) &= \sum_{i=1}^{m'} \hat{\mathrm{var}}(\Phi(X_v)^\top \mathbf{c}_i) \sum_{j=1}^{d} (\mathbf{c}_i^{(j)})^2 \\
&= \sum_{i=1}^{m'} \sum_{j=1}^{d} \hat{\mathrm{var}}(\Phi(X_v)^\top \mathbf{c}_i)(\mathbf{c}_i^{(j)})^2 \\
&= \sum_{j=1}^{d} \sum_{i=1}^{m'} \hat{\mathrm{var}}(\Phi(X_v)^\top \mathbf{c}_i \mathbf{c}_i^{(j)}) \\
&= \sum_{j=1}^{d} \hat{\mathrm{var}}(\sum_{i=1}^{m'} \Phi(X_v)^\top \mathbf{c}_i \mathbf{c}_i^{(j)}) \\
&= \sum_{j=1}^{d} \hat{\mathrm{var}}(P(\Phi(X_v),\mathbf{c})^{(j)}) \\
&= \hat{\mathrm{var}}(P(\Phi(X_v),\mathbf{c})).
\end{aligned}
\tag{8}
$$

The fourth equality holds since $\Phi(X_v)\mathbf{c}_i$ and $\Phi(X_v)\mathbf{c}_j$ are uncorrelated, which can be shown by calculating the co-variance between $\Phi(X_v)\mathbf{c}_i$ and $\Phi(X_v)\mathbf{c}_j$ as:

$$
(\Phi(X_v)\mathbf{c}_i - \hat{\mathbb{E}}_X[\Phi(X_v)^\top \mathbf{c}_i])(\Phi(X_v)^\top \mathbf{c}_j - \hat{\mathbb{E}}_X[\Phi(X_v)^\top \mathbf{c}_j])
$$

$$
= \sum_{t=1}^{d}(\Phi(X_v)^{(t)}\mathbf{c}_i^{(t)} - \hat{\mathbb{E}}_X[\Phi(X_v)^{(t)}\mathbf{c}_i^{(t)}]) \sum_{s=1}^{h}(\Phi(X_v)^{(s)}\mathbf{c}_j^{(s)} - \hat{\mathbb{E}}_X[\Phi(X_v)^{(s)}\mathbf{c}_j^{(s)}])
$$

$$
= \sum_{t=1}^{d}\sum_{s=1}^{d}\hat{\mathbb{E}}_X[\Phi(X_v)^{(t)}\Phi(X_v)^{(s)}]\mathbf{c}_i^{(t)}\mathbf{c}_j^{(s)} - \sum_{t=1}^{d}\sum_{s=1}^{d}\hat{\mathbb{E}}_X[\Phi(X_v)^{(t)}]\hat{\mathbb{E}}_X[\Phi(X_v)^{(s)}]\mathbf{c}_i^{(t)}\mathbf{c}_j^{(s)}
$$

$$
= \sum_{t=1}^{d}\sum_{s=1}^{d}\hat{\mathrm{cov}}(\Phi(X_v)^{(t)}, \Phi(X_v)^{(s)})\mathbf{c}_i^{(t)}\mathbf{c}_j^{(s)} = \sum_{t=1}^{d}\hat{\mathrm{var}}(\Phi(X_v)^{(t)})\mathbf{c}_i^{(t)}\mathbf{c}_j^{(t)} = 0.
$$

Where the last two equations follow by each dimension of $\Phi(X_v)$ is uncorrelated with unit variance and $\mathbf{c}_i$ and $\mathbf{c}_j$ is uncorrelated. By plugging in equation 8 into equation A, we may obtain

$$
\hat{\mathrm{var}}(P(\Phi(X_v), \mathbf{p})) \geqslant \hat{\mathrm{var}}(P(\Phi(X_v), \mathbf{c})).
$$

Define $\hat{\mathbf{c}}$ as the concatenated matrix for the orthonormal basis for orthogonal complement of $\mathbf{c}$, and define $\mathbf{c}_{all}$ by concatenating $\mathbf{c}$ and $\hat{\mathbf{c}}$. We know $\Phi(X_v) - P(\Phi(X_v), \mathbf{c}) = P(\Phi(X_v), \hat{\mathbf{c}})$ by fundamental properties of linear projections. Since all vectors in $\mathbf{c}$ is orthogonal to vectors in $\hat{\mathbf{c}}$ and by pluggin in equation 8 for $\mathbf{c} = \mathbf{c}_{all}$, we get $\hat{\mathrm{var}}(P(\Phi(X_v), \mathbf{c})) + \hat{\mathrm{var}}(P(\Phi(X_v), \hat{\mathbf{c}})) = \hat{\mathrm{var}}(P(\Phi(X_v), \mathbf{c}_{all})) = \hat{\mathrm{var}}(\Phi(X_v))$. By combining the observations we get

$$
\begin{aligned}
\hat{\mathrm{var}}(\Phi(X_v) - P(\Phi(X_v), \mathbf{c})) &= \hat{\mathrm{var}}(P(\Phi(X_v), \hat{\mathbf{c}})) \\
&= \hat{\mathrm{var}}(\Phi(X_v)) - \hat{\mathrm{var}}(P(\Phi(X_v), \mathbf{c})) \\
&\geqslant \hat{\mathrm{var}}(\Phi(X_v)) - \hat{\mathrm{var}}(P(\Phi(X_v), \mathbf{p})) \\
&= \hat{\mathrm{var}}(P(\Phi(X_v), \hat{\mathbf{p}})) \\
&= \hat{\mathrm{var}}(\Phi(X_v) - P(\Phi(X_v), \mathbf{p})).
\end{aligned}
$$

and following the isometry of D, we have

$$
\hat{\mathrm{var}}(h(\Phi(X_v) - P(\Phi(X_v), \mathbf{p}))) \leqslant \hat{\mathrm{var}}(h(\Phi(X_v) - P(\Phi(X_v), \mathbf{c}))),
$$

and thus the first m PCA vectors maximizes $\eta_2$. $\qquad\square$

## APPENDIX B  ADDITIONAL EXPERIMENTS RESULTS AND SETTINGS

**Detailed Experiment Settings in Toy Example**  The complete list of the target y is $y_1 =\sim (\xi_1 \cdot \xi_3) + \xi_4, y_2 = \xi_2 + \xi_3 + \xi_4, y_3 = \xi_2 \cdot \xi_3 + \xi_4 \cdot \xi_5, y_4 = \xi_2 \text{ XOR } \xi_3, y_5 = \xi_2 + \xi_5, y_6 =\sim (\xi_1 + \xi_4) + \xi_5, y_7 = (\xi_2 \cdot \xi_3) \text{ XOR } \xi_5, y_8 = \xi_1 \cdot \xi_5 + \xi_2, y_9 = \xi_3, y_{10} = (\xi_1 \cdot \xi_2) \text{ XOR } \xi_4, y_{11} =\sim (\xi_3 + \xi_5), y_{12} = \xi_1 + \xi_4 + \xi_5, y_{13} = \xi_2 \text{ XOR } \xi_3, y_{14} =\sim (\xi_1 \cdot \xi_5 + \xi_4), y_{15} = \xi_4 \text{ XOR } \xi_5$.

We create the dataset in matplotlib, where the color of each shape is sampled independently from green,red,blue,black,orange,purple,yellow, and the location is sampled randomly with the constraint that different shapes do not coincide with each other. For hyper-parameter selection, we simply set $\lambda_1 = \lambda_2 = 10.0$. We fix this hyper-parameter throughout all experiments to prevent exhaustive tuning and over-fitting. Scaling the hyper-parameter in the same order produces similar results. We use 1000 images in each cluster for all methods that are compared. For selecting the concepts in TCAV and ACE, we compare the number of labels where the concept has p-value < 0.16 and choose the top 5 concepts (since even TCAV score 1.0 does not have p-value < 0.05). We note that we have tried many alternatives for choosing concepts for TCAV and ACE, but failed to achieve better performance for TCAV and ACE. The main reason may be that the ground truth $y$ contains functional such as XOR, which has 0 TCAV score for inputs.

For a more concrete example, consider the case $Y = X_1 \text{ XOR } X_2$, and assume that we have 3 concepts candidates $X_1, X_2, X_3$. All 3 concepts would have 0 TCAV score when each concept has independent Bernoulli distribution with $p = 0.5$. Therefore, TCAV will choose $X_1, X_2, X_3$ with an equal probability. Although our method also produces a linear concept direction, the completeness measure for $\{X_1, X_2\}$ would be 1, while the completeness measure for completeness measure for $\{X_1, X_3\}$ and $\{X_2, X_3\}$ would be far less than 1. The reason is that we project the activation space onto the concept space, and then pass through the remaining model to get our result. By projecting the activation space onto the span of $\{X_1, X_2\}$, we can still get $Y = X_1 \text{ XOR } X_2$. On the other hand, if we project the activation space onto $\{X1, X_3\}$, the information of $X_2$ would be loss, and thus we get a much worse completeness score. The key difference between our method and TCAV is that we feed the projected activations back into the original model (which is $h(.)$ in our problem setting), which may capture the non-linear relationship between the projected space and the output. Such a non-linear relationship might be neglected in the TCAV score.

**Implementation Details**  For calculating ConceptSHAP, we use the method in kernelSHAP (Lundberg & Lee, 2017) to calculate the Shapley values efficiently. Before calculating the nearest neighbor, we ensure that the dot product between each concept vector and its most salient cluster mean has a positive dot product (if it is negative, we take the negative of the concept vector as the new concept vector, which does not effect the loss at all). For the input cluster proposals in AwA, we follow the code of **?** and their hyper-parameters. For input cluster in Imdb, we obtain 500 clusters from positive sub-sentences and 500 clusters from negative sub-sentences by k-means clustering. We train a linear classifier differentiating the cluster segments and random segments, and remove clusters with accuracy lower than 0.95. We also remove clusters that have less than 100 elements. For input cluster proposals in the toy dataset, we used k-means clustering with 20 clusters.

**Additional Nearest Neighbors for IMDB**  We show addition nearest neighbors for each concept obtained in IMDB in Figure 5. We observe that some top nearest sub-sentences of concept 2 and concept 3 do not have the word "think" in it. The top nearest neighbors in concept 4 generally has a tone that encourages readers of the review to watch the movie, which is probably why it has the largest ConceptSHAP score.

**Additional Nearest Neighbors for AwA**  We show addition nearest neighbors for each concept obtained in AwA in Figure 6. The nearest neighbors all share the same texture. Interestingly, some of the nearest neighbors of ripple 2 are not exactly ripple, but tree/leaves that share similar texture as ripple. Some nearest neighbors of dots contains dots from leaves instead of pure dots on animals. This again validates that the concepts are based on the texture of the image.

**Concept 1**

characters are fun to watch and you can see the
decided to actually watch this film i found the characters
well fire against ice with characters named nekron and darkwolf
and passion for the future the characters of jesse hawke
different and i thought the main characters suspiciously dressed like
characters ranma you would think making him too strong would
characters for your enjoyment only watch it let go and
neverending immoral relationships of the show's characters everybody seems to
kurosawa weaves a tale that has a cast of characters

**Concept 2**

that keeps on reappearing to the scene where you think
she deserved a more studied finale than that i think
think no sometimes hatred and isolation are deeper are more
think established the trade in african slaves in the first
that has produced such excellent tv comedies seems to think
think that by having this man's face on their chest
that has since fell into disrepair and who would think
will pull you in make you laugh make you think
are sometimes entertaining by virtue of their very datedness flared

**Concept 3**

i think the most frustrating thing is that the performances
you might think to see organs yanked out of the
noise or when he eventually kills him by scraping his
many people think has an underlying meaning the love between
i'll admit i think uma thurman is the most beautiful
like to think i have a better grasp on asian
i think they were proper actor's rather than friends or
son zizek may object that she also evidently enjoys rough
also when i watch gunga din i think of star

**Concept 4**

make and its nice for your kids to watch and
this movie and it becomes something you can watch without
have to type that geek because i'd totally watch this
ever made all of you must watch this perfection 10
ok then my friends at work said watch it again
into play and he is never boring to watch he
what kind of movie you're about to watch father makes
also forced 4 of my friends to watch it with
around in your head you will want to watch this

**Concept 5**

soon after their version of the film was made and
paid so well after all acting is one of the
soon after watching this film you will realize why it
right after this little girl killed the first person very
america for me was dreading watching this after all the
it after watching it you will say that it was
an enjoyable movie after an opening that i would best
the film again after several years without seeing it this
again after watching the 1 1 2 i was like

Figure 5: Additional Nearest Neighbors for each concept obtained in IMDB.

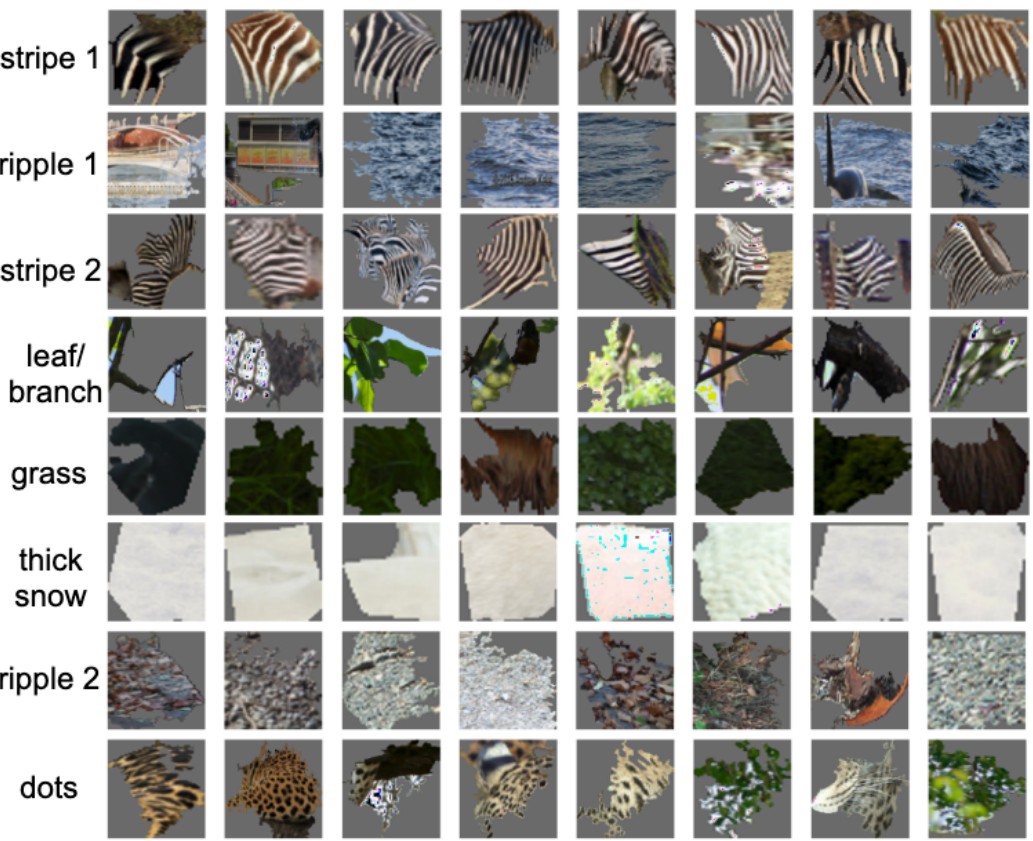

Figure 6: More Nearest Neighbors for each concept obtained in AwA.

## APPENDIX C    ADDITIONAL EXPERIMENTS

Throughout this section, we refer to $\eta^{(1)}$ when the term completeness score is mentioned for the simplicity of presentation.

### C.1    HYPER-PARAMETERS

We run a controlled experiment on different $\lambda = \lambda_1 = \lambda_2$ values for our concept discovery algorithm to show the robustness of our algorithm against different hyper-parameters. We also perform an ablation study when we do not optimize for the completeness score, which we call ours_unsup* and ours_sup*. We summarize the result in Figure 7. We observe that our method outperforms the baselines for $\lambda \in [0.5, 50]$. This shows that our concept discovery algorithm is robust with respect to the hyper-parameter $\lambda$. We also observe that for the controlled version where the completeness score is not optimized, the alignment score and completeness score is worse compared to the baselines. This shows that the completeness objective is essential in obtaining complete and correct concepts in our algorithm.

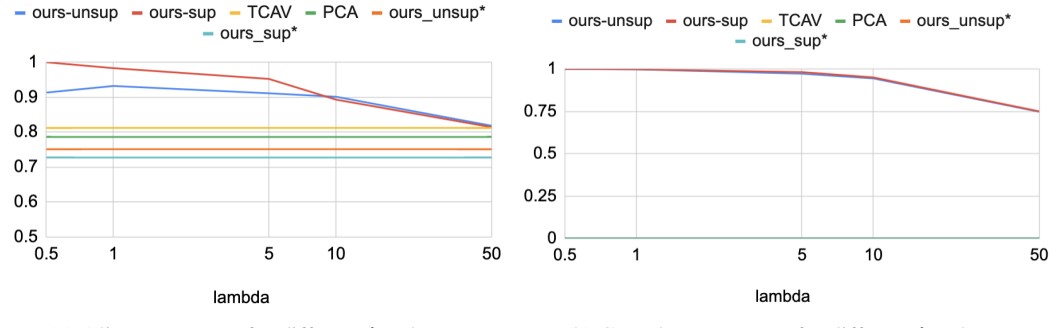

(a) Alignment score for different $\lambda$ values.          (b) Completeness score for different $\lambda$ values.

Figure 7: The alignment score and completeness score for our method with different hyper-parameter for the toy dataset.

## C.2 NUMBER OF CONCEPTS

We show the performance metric when different number of concepts are retrieved for both our concept discovery algorithm and the baselines in the toy dataset as well as the AwA dataset respectively in figure 8 and 9. In the toy dataset, we observe that the completeness for all the baselines are low compared to the completeness of our method. On the other hand, the alignment score of our method increases more significantly compared to the baselines when the number of concepts increases from 1 to 9. Only ours-sup and ours-unsup achieves a high completeness score when the concept number is larger or equal to 4, which may explain the superior performance of our method on the alignment score. As we have argued in the introduction, concepts that are not complete fails to "fully" interpret the model, and thus since most baseline methods are not complete, they fail to reach a high alignment score. On the other hand, our method which obtains a high completeness score achieves a high alignment score.

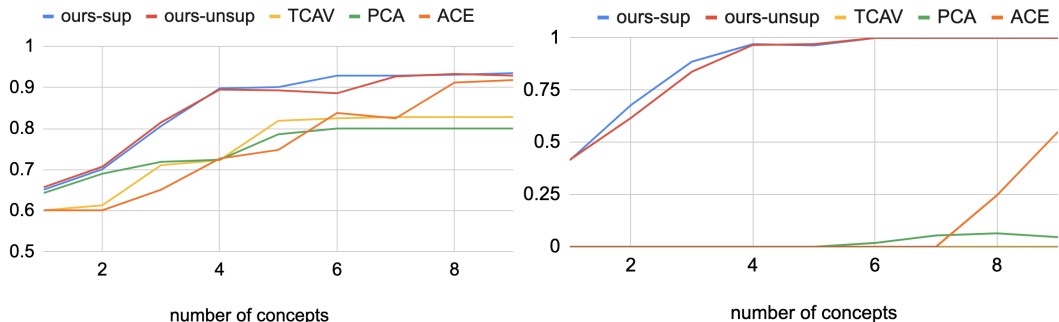

(a) Alignment score for different number of concepts. (b) Completeness score for different number of concepts.

Figure 8: The alignment score and completeness score for all methods with different number of discovered concepts are chosen for the toy dataset.

For the AwA dataset, we only show the completeness score for different algorithms with increasing number of concepts. We observe that the completeness of our method is much higher than ACE and PCA. We note that PCA still achieves satisfactory completeness score when the number of concepts is high. This is not surprising since we proved in Proposition 2.1 that PCA maximizes the completeness under isometry of the model. We point out that a larger number of total concepts makes the interpretation more understandable by human, and thus more interpretable. However, the explanation obtained by PCA is only close to complete (i.e. has a completeness larger than 0.95) with 29 concepts. This experiment provides additional empirical support that PCA obtains worse completeness score compared to our method, which may only be caused that the isometry (and perfect accuracy) assumption not holding in practice. This further validates the effectiveness of our method against naive PCA.

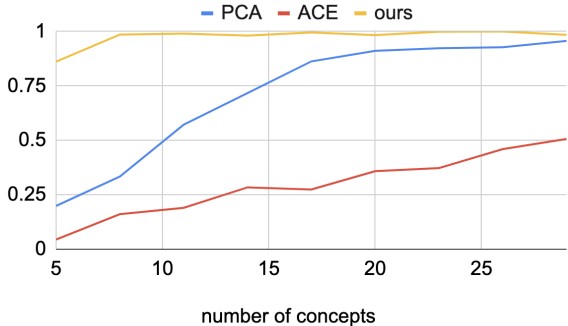

Figure 9: Completeness score for different number of concepts in AwA.

## C.3   ROBUSTNESS AGAINST NOISE

We would also be interested in how the completeness metric may change when the input is slightly perturbed. We perform an experiment on AwA where a Gaussian random noise is added to all the input in the validation set. We plot the completeness score against different standard deviation for each dimension of the added noise in Figure 10. We observe that the completeness score is still above 0.96 even when a multivariate Gaussian noise with standard deviation 10 in each dimension is added to the original input for all the validation set data. We further note that since the completeness score is calculated by all the validation data, perturbing only one validation point will cause negligible impact on the completeness score.

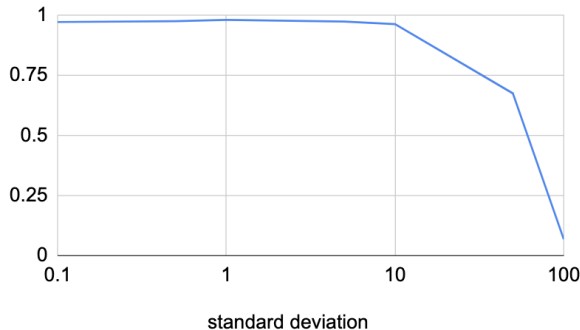

Figure 10: Completeness score for different standard deviation of added noise in AwA.

## C.4   OPTIMIZING WITHOUT COMPLETENESS

We conduct an ablation study on how the result changes when we drop the completeness term in the main objective. The results for optimizing without completeness in the toy dataset has been shown by ours_unsup* and ours_sup* in section C.1, which is much worse compared to ours with the completeness score optimized. We further show results of the concepts when optimizing without the completeness metric. We note that the completeness score is 0 for AwA dataset. We further visualize the top nearest neighbors for the concepts that are optimized without the completeness metric in Figure 11. We observe that while some concepts are the same as ours when the completeness is optimized, it contains more concepts on wheels and bars which does not seem to be very related to the classification of animals. We notice that the top nearest neighbors are still coherent in general. This is not surprising since our cluster-sparsity regularizers enforces the discovered to be coherent, however, the result now may not be sufficient to models prediction since we dropped the completeness objective and the completeness score becomes 0.

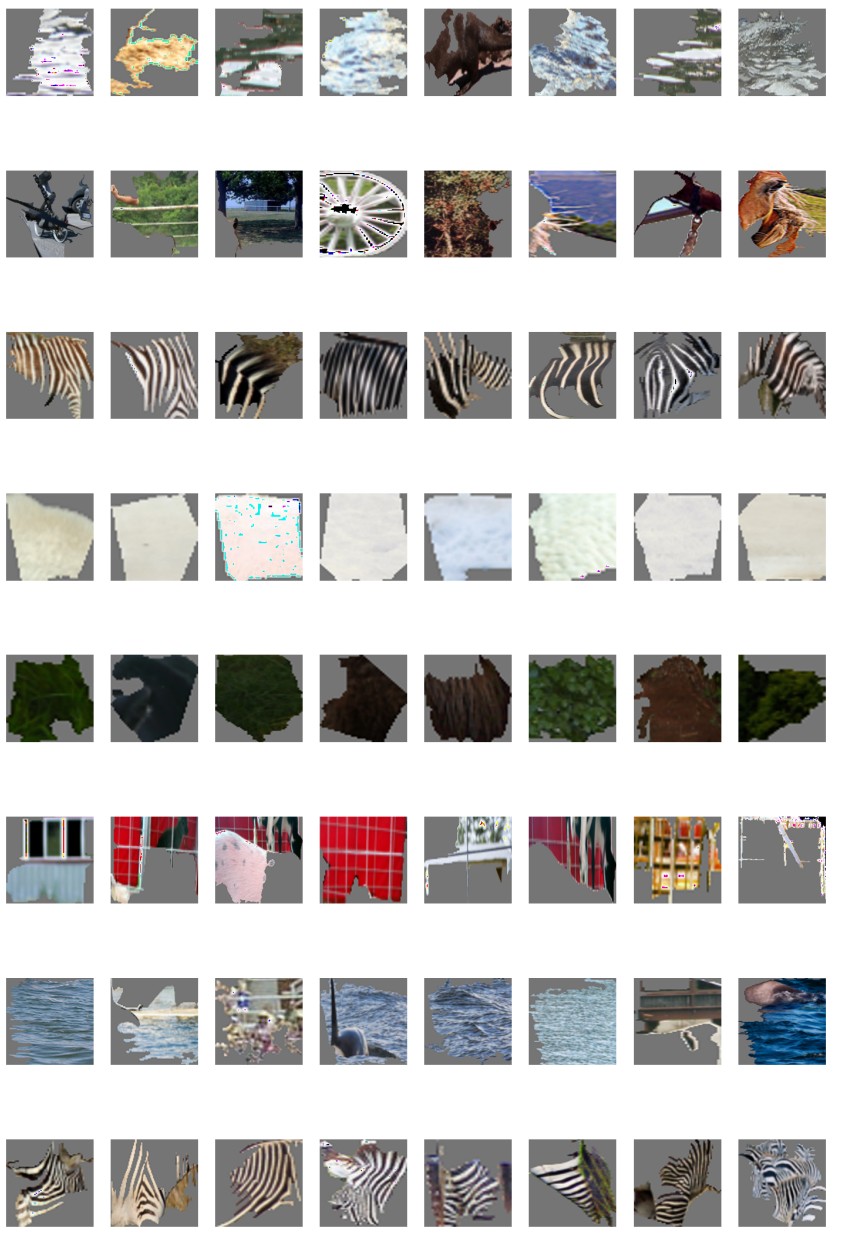

Figure 11: Nearest Neighbors for each concept obtained in AwA without the completeness metric.

