# OpenReview forum: "On Concept-Based Explanations in Deep Neural Networks"
_ICLR.cc/2020/Conference — Reject_

### Official Review · AnonReviewer1 · 2019-10-23
**Official Blind Review #1**

**Rating:** 6

**Review:**

Summary

The paper proposes metrics for evaluating concept based explanations in terms of ‘completeness’ -- characterized by (1) whether the set of presented concepts if sufficient to retain the predictive performance of the original model and (2) how is performance affected when all information useful to a complete set of concepts (as per (1)) is removed from features at a specific layer. Assuming concept vectors lie in linear sub-spaces of the activations of the network at a specific layer, the underlying assumption is that if a given set of ‘concept’ vectors is complete, then using a projection of the intermediate features from input onto the sub-space spanned by concepts should not result in reduced / affected predictive performance. Based on these characterizations, the paper proposes an objective to discovering complete and interpretable set of concepts given a candidate cluster of concepts. Furthermore, the paper proposes metrics to quantify the importance of each concept (using Shapley values) and per-class importance of concepts. The authors conduct experiments on toy data, image and text classification datasets and show that their proposed approach can discover concepts that are complete and interpretable.

Strengths

- The paper is well-written and generally easy to follow. The authors do a good job of motivating the need for the completeness metric, characterizing it specifically in the case of concepts spanning sub-spaces of activations and subsequently utilizing the same to motivate an effective concept discovery method.

- The proposed approach and angle being looked at in the paper is novel in the sense that while prior work has mostly focused on characterizing concepts which are salient. Ensuring that concepts are sufficient for predictive performance ensures the fidelity of the interpretability approach.

- I like the fact that the authors decided to capture both aspects of the completeness criterion -- (1) projection to concept-space should not hurt performance and (2) how does removing concept-projected information from the features affect performance. Capturing both provides a holistic viewpoint of the features of the concerned layer -- (1) can explain features/decisions with an associated metric based on the ‘imperfect’ set of concepts and (2) captures the effectiveness of the information present in the features if we remove all concept-useful information.

- The choice of using Shapley values in ConceptSHAP as a metrics provides a whole range of desirable properties in the metrics used indicate the quality of concepts at different levels of granularity -- per-class importance of concepts across classes adding up to the overall importance of a concept. Furthermore, the observation that under certain conditions, the top-k PCA vectors maximize the defined completeness scores is interesting as well.

Weaknesses

Having said that, I’m interested to hear the thoughts of the authors on the below two points. My primary (not major) concern is the fact the proposed approach is only centered around ensuring and evaluating fidelity of the discovered concepts to the original model.

- While the proposed approach and evaluation metrics are novel, and the results generally support the claims of the paper -- toy experiments result in recovery of the ground-truth concepts, concepts identified for text and image classification offer feasible takeaways -- there is still a lack of proper human-interpretability aspect of the discovered concepts. The proposed approach to discover complete concepts mostly acts similar to a pruning approach on top of a candidate set of concepts based solely on fidelity to the original model. The obtained concepts are mostly explained via feasible hypotheses. One possible experiment that can be used to capture the reliability aspect of the discovered concepts could be as follows -- “Given the set of concepts (and representative patches) and SHAP values across all (or most relevant) classes, are humans able to predict the output of the model?” Is it possible to setup and experiment of this sort? I believe it might help understand the utility of the SHAP values in this context (beyond the advantages in terms of manipulation and characterized completeness).

- The experimental results for image classification are presented on the Animals with Attributes (AwA) dataset. AWA is a fine-grained dataset with only one class present per-image. I’m curious to what happens when the same approach is applied to datasets where images have multiple classes (and potentially distractor classes) present. Is it possible that it becomes harder to discover ‘complete’ concepts (subject to the availability of a decent approach to provide an initial set of candidate clusters). Do the authors have any thoughts on this and any potential experiments that might address this?

Reasons for rating

Beyond the above points of discussion, I don’t have major weaknesses to point out. I generally like the paper. The authors do a good job of identifying the sliver in which they make their contribution and motivate the same appropriately. The proposed evaluation metrics and discovery objectives offer several advantages and can therefore generally serve as useful quantifiers of concept-based explanation approaches. The strengths and weaknesses highlighted above form the basis of my rating.


**Experience Assessment:**

I have read many papers in this area.

**Review Assessment: Checking Correctness Of Derivations And Theory:**

I assessed the sensibility of the derivations and theory.

**Review Assessment: Checking Correctness Of Experiments:**

I assessed the sensibility of the experiments.

**Review Assessment: Thoroughness In Paper Reading:**

I read the paper thoroughly.

---

> ### Author Response · Authors · 2019-11-13
> **Thanks for the constructive review**
>
> We thank the reviewer for the positive and constructive comments.
>
> -- One possible experiment that can be used to capture the reliability aspect of the discovered concepts could be as follows -- “Given the set of concepts (and representative patches) and SHAP values across all (or most relevant) classes, are humans able to predict the output of the model?”
>
> Reply: We certainly agree that human experiments would greatly strengthen our paper, and we hope to include human experiments in our future work. We think they are beyond the scope of this paper because they are non-trivial to design properly.
>
> -- Multi-label dataset: I’m curious to what happens when the same approach is applied to datasets where images have multiple classes (and potentially distractor classes) present. Is it possible that it becomes harder to discover ‘complete’ concepts (subject to the availability of a decent approach to provide an initial set of candidate clusters).
>
> Reply: We agree that multi-label classification would be more difficult for our concept discovery algorithm is most cases. However, we would like to point out that our toy example is indeed a multi-label dataset. Each input has a 15-dimension binary label, and each label depends on a subset of the 5 ground truth concepts (where the detailed function form is listed in Appendix B.) We hope to include a more difficult multi-label dataset in the future to verify whether our method can still work well.

---

> > ### Comment · AnonReviewer1 · 2019-11-15
> > **Thanks for responding to the comments.**
> >
> > Thanks to the authors for responding to my comments.
> >
> > >We agree that multi-label classification would be more difficult for our concept discovery algorithm is most cases. However, we would like to point out that our toy example is indeed a multi-label dataset. Each input has a 15-dimension binary label, and each label depends on a subset of the 5 ground truth concepts (where the detailed function form is listed in Appendix B.) We hope to include a more difficult multi-label dataset in the future to verify whether our method can still work well.
> >
> > Thanks for responding to this and pointing to the appropriate analogous setting.

---

### Official Review · AnonReviewer2 · 2019-10-24
**Official Blind Review #2**

**Rating:** 3

**Review:**

The authors build on the work by Ghorbani et al. in concept-based interpretability methods by taking into account the "completeness" of the concepts. This basically tests whether the models accuracy holds if the input is projected onto the span of the discovered concepts. They propose "ConceptSHAP", based on Shapley values, to assign importance to the learned concepts. These could be shown to satisfy some reasonable properties such as efficiency (sum of importances equals total completeness value), symmetry, additivity, dummy (a concept that does not change the completeness universally should have zero importance) as stated in Prop. 4.1. The method is finally tested on a variety of datasets, including a synthetic one, which shows that optimizing "completeness" helps in discovering a richer variety of important  concepts than prior work).
- I was wondering how the completeness and importance measures change when the input is perturbed slightly such that the classifier output doesn't change?
- How the concepts would change if the completeness measure is removed from the optimization in text and image-classification?

**Experience Assessment:**

I have read many papers in this area.

**Review Assessment: Checking Correctness Of Derivations And Theory:**

I assessed the sensibility of the derivations and theory.

**Review Assessment: Checking Correctness Of Experiments:**

I assessed the sensibility of the experiments.

**Review Assessment: Thoroughness In Paper Reading:**

I made a quick assessment of this paper.

---

> ### Author Response · Authors · 2019-11-13
> **reply to your questions**
>
> We address the problems listed by the reviewers via additional experiments.
>
> -- I was wondering how the completeness and importance measures change when the input is perturbed slightly such that the classifier output doesn't change?
>
> Reply: We have added a new section, Appendix C.3, where we perform an experiment on AwA by adding a Gaussian noise to all the inputs in the validation set. We show the completeness score vs. noise level in Figure 10. We observe that the completeness score is still above 0.96 even when a multivariate Gaussian noise with standard deviation 10 in each dimension is added to the original input for all the validation set data. This shows that the completeness measure is robust against slight and even non-negligible random perturbations on the validation set. We further note that since the completeness score is calculated on all the validation data, perturbing only one validation point will cause negligible impact on the completeness score.
>
> -- How the concepts would change if the completeness measure is removed from the optimization?
>
> Reply: We conduct additional experiments in Appendix C.4 and Appendix C.1, to show the results when the completeness measure is removed from the optimization for AwA and the toy dataset. In both cases, we see that the completeness score is decreased to 0. We also observe that the alignment score is decreased to perform worse than the baseline in the toy dataset, and the nearest neighbors in AwA seems to contain less relevant concepts to animals. We argue that such an ablation study further validates the effectiveness of the completeness score in obtaining satisfactory concepts.
>
> Since the reviewer did not point out any weak points of paper, we sincerely hope the reviewer to consider raising the score after we addressed the reviewer’s questions.

---

### Official Review · AnonReviewer3 · 2019-11-04
**Official Blind Review #3**

**Rating:** 3

**Review:**

The authors are concerned with improving algorithms that propose sets of
concept vectors to help explain the prediction of a pre-trained network on a
held-out test examples. The paper examines "completeness" (Gilpin et al 2018) of
the context vector set---projecting activations on to the concept subspace does
not hurt predictive performance---as a desired criteria that was overlooked by
previous concept-based explanation methods. They show that PCA on activations
yields complete concepts under some bijectivity assumption on the final layers
of the network. They then discuss how to produce a set of concept vectors that
maximize completeness under sparsity priors when supervised or unsupervised
clusters of candidate concepts are given. They also show that SHAP values can be
applied to compute the relative importance of each concept to the overall
completeness score.

I find the focus of this research---on the completeness of concept
explanations---to be quite interesting and relevant to the interpretable machine
learning literature. The authors have approached this question from several
directions and have demonstrated a fluency with the existing literature and its
limitations. However, I think they could do much more to motivate their
proposals by convincing the reader that (a) existing concept-based explanations
are not complete in practice and (b) PCA in the activation space is not
sufficient as a baseline method for proposing concepts. In the current revision,
the paper lacks coherency---the proposals in each section do not seem connected
to one another---and it is difficult to assess from the experiments whether the
proposed algorithms address a known problem in existing methods.

For example, the propositions in S2 rely on the ability of the post-concept
function h to bijectively map between the Frobenius norm in the concept space to
the loss function in the logit space. Therefore it is logical to ask whether
concepts can be found directly via PCA on the concept space. The authors
speculate that this bijectivity of h is unlikely to hold in practice, and
therefore PCA would be insufficient. However I would have been more convinced by
some empirical evidence for this claim. What about the case where f is bijective
(i.e. an invertible neural network); is PCA on the activation space sufficient
to produce complete and useful concepts in
this case?

I also wondered whether there is a corner case where the functional form of h
causes the components that capture the most variance in the activation space
(i.e. top PCA vectors) to capture the least variance in the logits space. This
seems plausible if h is an unconstrained linear mapping or non-linear mapping;
think about a simple whitening function that scales principal components in
inverse proportion to their eigenvalue. In this case we may be better off using
the bottom PCA vectors than the top ones. Admittedly this is only thought
experiment, but is there any way for the authors to show that this does not happen
in practice by characterizing the h functions we see for neural networks applied
to application areas of interest: sentiment analysis and image classification?

The synthetic experiments seem useful to the overall story (although the data
generating process is somewhat hard to parse at first), and I like that
there are many baseline methods in this study. It seems that PCA is neither
complete nor aligned in this simple setting. It also seems that TCAV is not
complete. Unfortunately the authors fail to show that similar trends hold in the
non-synthetic setting. My feeling is that a more comprehensive and thorough
experimental study---starting from a characterization of a problem with existing
methods---would strengthen the paper considerably.


Some superficial comments:
S1
* "This has thus lead to an increasing interest..." It would be good to add
  references to this claim.
* "Of course such degeneracy assumptions likely not hold" -> "Of course such
  degeneracy assumptions likely do not hold" ->
* "...which can explain how much does each concept contribute to the
  completeness score" -> "...which accounts for the contribution of each concept
  to the completeness score..."
S2
* "...that capture how complete is a given set of given concepts" -> "...that
  capture the completeness of a given set of given concepts"
* "...concepts hold sufficient info for prediction" -> "...concepts hold
  sufficient information for prediction"
* "...prediction scores for examples in class A won't be much different from
  examples in class A" -> "...prediction scores for examples in class A won't be
  much different from other examples in class A"?
S3
* "...each concept direction is semantically meaningful to human" -> "...each
  concept direction is semantically meaningful to humans"
S5
* "y1 = ~..." could you make a note that "~" denotes logical not?



**Experience Assessment:**

I have published one or two papers in this area.

**Review Assessment: Checking Correctness Of Derivations And Theory:**

I assessed the sensibility of the derivations and theory.

**Review Assessment: Checking Correctness Of Experiments:**

I carefully checked the experiments.

**Review Assessment: Thoroughness In Paper Reading:**

I made a quick assessment of this paper.

---

> ### Author Response · Authors · 2019-11-13
> **Thanks for the constructive comments**
>
> We thank the reviewer for the constructive comments. We address the questions raised by the reviewer in the following:
>
> -- Convince readers that existing concept-based explanations are not complete in practice and PCA in the activation space is not sufficient as a baseline method for proposing concepts.
>
> Reply: We agree that giving empirical evidence on existing concept-based explanations are not complete in practice would strengthen the paper considerably. We provide the completeness score of our method and TCAV, ACE, PCA in the toy dataset and AwA in Appendix C.2 with varying number of concepts being retrieved. In figure 8, we see that TCAV, PCA, ACE are not complete even when retrieving 9 concepts, and thus have a worse alignment score. In figure 9, we see that the completeness of our method is much higher than ACE and PCA in AwA. Our concept when retrieving 8 concepts has a higher completeness score than both PCA and ACE when retrieving 29 concepts. We believe these are strong evidence to show that existing concept explanations are not complete, and thus may not be sufficient to explain the model fully. For further interpretation of the experiments please see the descriptions in Appendix C.2.
>
> We would point out that the difficulty to evaluate whether TCAV is complete generally is that TCAV relies on concepts that are defined by humans. Therefore, one can only point out that TCAV in a certain dataset with certain concepts are not complete, but additional labeling by humans may make TCAV complete. However, since the completeness metric can be used to evaluate whether a set of concepts obtained by TCAV are complete or not, it may be used to improve the practical usage of TCAV. In this paper, we mainly argue that verifying whether a set of concepts is complete is an important property that is overlooked by previous concept-based explanations. We also show that for some examples current concept-based explanations are not complete and thus are not sufficient to fully explain the model. Such a rigorous criteria would be a good sanity check for both existing concept explanations and future concept explanations.
>
> -- empirical evidence on whether PCA would be sufficient to discover concepts
>
> Reply: In Appendix C.2, we see that PCA has much worse completeness score compared to our method. Since the classifier in the toy example and AwA both have a high classification accuracy, the gap between the completeness of PCA and our method should be due to the non-isometry of the network. We also would like to emphasize that under Proposition 2.2, we highlight the main difference between PCA and our method. While we show that our method achieves higher completeness compared to PCA, we also include the cluster-sparsity regularizer to make each concept more interpretable to humans, which is not achieved by PCA as it greedily maximizes the reconstruction error for each concept dimension.

---

> ### Author Response · Authors · 2019-11-15
> **Additional Response**
>
> We would like to point to our response to Reviewer4, which gives an example why TCAV score may not capture the set of useful concepts. This is another limitation of existing methods (for TCAV and ACE) compared to the completeness score we propose. We will add these discussions in the final draft to improve our paper. We also addressed the additional (superficial) comments by the reviewers in our updated version.

---

### Official Review · AnonReviewer4 · 2019-11-05
**Official Blind Review #4**

**Rating:** 6

**Review:**

Update: after reading the rebuttals, I raised my rating to weak accept. Authors provide an interesting and possible limitation of TCAV during rebuttal (in the XOR case TCAV might not be able to identify meaningful subspace). It would strengthen the paper by somehow proving it. Maybe you can visualize the TCAV vectors and your vectors in the simulation and use it to explain why TCAV selects unrelevant concept 11, or designs another simple experiments.
Also, if accepted, please consider more parameter stability analysis in the real datasets, including ablation studies of different regularizations. This helps readers understand the value of this method in the real data.

Overall I like the paper much better after rebuttal, but still have a lukewarm feeling about it. I do not see lots of value in the current method, probably due to not very interesting or interpretable results in the real data. However, it does provide a new concept of completeness, and I agree with R1 that it's interesting and relevant. So I would change my rating to weak accept.

====================================================================================================
Original evaluation:

This paper can be seen as an extension to the paper Ghorbani et al., 2019 to try to extract "complete" unsupervised concepts as well as maintainting interpretability. Compared to Ghorbani et al., they have 2 modifications. First, they try to learn a set of concept vectors such that the resulting projected activations on this concept basis would not lose too much accuracy, which they call completeness. Second, to maintain interpretability for these vectors, they regularize these vectors to sparsely match to some input clusters (generated by the k-means) that human can examine, and regularize no two concept matches to the same cluster. They validate their concepts in the simulation, text and image datasets.

Overall I feel the idea is interesting but a bit straightforward. The resulting concept vectors are also not very interpretable or interesting in the text and image domain, which seems to limit its utility to practioners.

Also, one of the problem is how to select the coefficients for the regularization. In the supplements authors mention that they choose lambda=10 and it produces "reasonable" results in the simulation. This decreases the credibility of the simulation that shows it outperforms the baselines such as TCAV, as it sounds like you set the parameters to get good metrics on this simulation. Moreover, in the real-world data there might not be an easy way to set these parameters. I think the authors could strengthen the paper by experimenting with different coefficients and examine the stability of the resulting explanations, or at least indicate how results differ if the coefficients are set differently. Or do an ablation study about how different objectives shape the resulting concepts.

Besides, I will suggest to plot a figure to show number of concepts used v.s. the completeness score. It will motivate why choosing specific number of concepts in the experiements. Also experimenting with different number of concepts is helpful for readers to see the stability of this method as well.

Some minor questions:
1. The claim of the result of the image dataset says it focuses on the texture. I think it might be caused by the super-pixel segmentation, which already removes most of the edges and leaves with the texture clusters.
2. Any intuition of why TCAV performs worse in the simulation? How do you select the top 5 vectors for the TCAV, and why does it select non-useful concept 11?
3. In the definition saliency score (page 4), I am just wondering if the normalized vector could be better? So the resulting value would not be affected by different magnitude of u_k or concept vectors c_j.
4. In the definition of the conceptSHAP score, shouldn't it be the average of conceptSHAP score across all the classes instead of the sum (inline equation near eq. 6)? If you have 1000 class, then the conceptSHAP score could be something around 1000, that will be wierd.

Writing:
- I am confused at first read in the 2nd paragraph of the section 3 in page 4. The word "concept" is overloaded and represent both as input concept clusters and the resulting concept vectors. Such as "...candidate clusters of concepts..." or "... each concept is salient to one cluster ...".
- Typo: "They main takeaway" -> "The main takeaway"

Strengths:
+ Good related work
+ Somewhat complete evaluation
Weaknesses:
- No analysis with so many hyparparameters (reg lambda, number of concepts), and thus not sure about the validity of the simulation
- Idea is interesting but straightforward
- Not very interpretable results

**Experience Assessment:**

I have published one or two papers in this area.

**Review Assessment: Checking Correctness Of Derivations And Theory:**

I assessed the sensibility of the derivations and theory.

**Review Assessment: Checking Correctness Of Experiments:**

I carefully checked the experiments.

**Review Assessment: Thoroughness In Paper Reading:**

I read the paper thoroughly.

---

> ### Author Response · Authors · 2019-11-13
> **Thanks for the constructive comments**
>
> We thank the reviewer for the constructive comments. We address the questions raised by the reviewer in the following:
>
> --idea is interesting but a bit straightforward.
>
> Reply: We believe that a simple elegant solution that is based on theoretical foundations and that yields promising results in an early-developing field of concept-based explainability, should be more valuable in providing useful insights compared to a complicated method.
>
> -- robustness of methods for different coefficients for the regularization and how to tune the hyper-parameter
>
> Reply: We set the regularization strength to a fixed value to avoid extensive tuning of our method. To show how our method performs when the regularizer strength varies, we conduct additional experiments in Appendix C.1 for the toy example. We vary the regularizer $\lambda$ from 0.5 to 50, and we show that the model performs well for $\lambda <=10$. This provides evidence that our promising results are not based on extensive hyper-parameter tuning. In fact, our method gives satisfactory performance for $\lambda$ between 0.5 and 10, which shows that our method is quite robust against different hyper-parameter values.
>
> In general, one could tune the regularization term by simply choosing the strongest regularization strength which gives a satisfactory completeness score in real world data where the correctness of the explanations cannot be evaluated.
>
> -- performance of all methods for different number of concepts
>
> Reply: We provide the completeness score of our method and TCAV, ACE, PCA in the toy dataset and AwA in Appendix C.2 with varying number of concepts being retrieved. In Figure 8, We see that our methods reaches a high alignment score when retrieving more than or equal to 4 concepts. This is not surprising since there are 5 ground truth concepts when constructing the dataset. In figure 8, we see that TCAV, PCA, and ACE do not yield high completeness score even when retrieving 9 concepts, and thus have a worse alignment score compared to our method. In Figure 9, we see that the completeness of our method is much higher than ACE and PCA in AwA. Our concept when retrieving 8 concepts has a higher completeness score than both PCA and ACE when retrieving 29 concepts. For further interpretations of Figure 9, please refer to Appendix C.2. In general, we believe that our method is stable with respect to the number of concepts retrieved.
>
> -- super-pixel segmentation removes most of the edges and leaves with the texture clusters
>
> Reply: We agree that the super-pixel segmentation may be one of the reasons that the retrieved concepts contains texture. However, in Appendix C.4 we find that when optimizing without the completeness, the retrieved concepts contains top nearest neighbors that does not share similarity in texture (see the second row and the sixth row). Therefore, this shows evidence that texture information is still important for the concepts to be complete to the model prediction.
>
> -- intuition of why TCAV performs worse in the simulation
>
> Reply: In “Detailed Experiment Settings in Toy Example”, we provide intuitions on why TCAV performs worse in the toy example. We give a concrete example for one potential reason why TCAV may fail in the toy example: consider the simple function Y = X_1 XOR X_2, for Bernoulli distributed binary X_1 and X_2 with p = 0.5. The TCAV score for X_1 and X_2 will both be zero in this case, which is not meaningful. TCAV may produce concepts that are not useful since it only captures the first order information between concepts and output, while the concept and output may have relationships that cannot be captured by first order methods.
>
> --the average of conceptSHAP score across all the classes instead of the sum
>
> Reply: We point out that the per-class conceptSHAP in Eq.(5) and Eq. (6) is defined to be around 1/(number of classes). Therefore, the total conceptSHAP for 1000 class would be around 1000* 1/1000, which is around 1. We may alternatively normalize Eq. (5) and Eq. (6) to around 1 and change the total conceptSHAP as an average of the per-class conceptSHAP. We note that the relative scaling does not make a difference since we only show the most important concepts (based on conceptSHAP with respect to this class) for each class.

---

> > ### Comment · AnonReviewer4 · 2019-11-14
> > **Minor clarification**
> >
> > Your response above says "TCAV may produce concepts that are not useful since it only captures the first order information". It sounds like you're talking about the limitation of linear concept vector. Then doesn't your method also suffer from the same drawbacks since your concept vectors are also assumed linear?

---

> > > ### Author Response · Authors · 2019-11-14
> > > **further clarifications**
> > >
> > > Thanks for the question, we hope to further clarify this and we will add the clarifications into our manuscripts in the future. Consider the case Y = X_1 XOR X_2, and assume that we have 3 concepts candidates X_1, X_2, X_3. All 3 concepts would have 0 TCAV score when each concept has independent Bernoulli distribution with p = 0.5. Therefore, TCAV will choose X_1, X_2, X_3 with an equal probability.
> > >
> > > Although our method also produces a linear concept direction, the completeness measure for {X_1, X_2} would be 1, while the completeness measure fo  completeness measure for {X_1, X_3} and {X_2, X_3} would be far less than 1. The reason is that we project the activation space onto the concept space, and then pass through the remaining model to get our result. By projecting the activation space onto the span of {X_1, X_2}, we can still get Y = X_1 XOR X_2. On the other hand, if we project the activation space onto {X1, X_3}, the information of X_2 would be loss, and thus we get a much worse completeness score. The key difference is that we feed the projected activations back into the original model ( which is h(.) in our problem setting), which may capture the non-linear relationship between the projected space and the output. Such a non-linear relationship might be neglected in the TCAV score.
> > >
> > > The main difference is that TCAV assumes a first order relationship exists between the output and the concept, while our method only assumes that concepts lie in linear direction in the activation space (without assuming the relationship between concepts and the output can be captured by the first order relationship).

---

### Author Response · Authors · 2019-11-13
**Summary of additional experiments**

We have added additional experiments to Appendix C to address the questions raised by the reviewers:
-- In Appendix C.1, we show the performance of our method with different hyper-parameters for the toy example, and the performance is quite stable when $\lambda$ is between 0.5 and 10. We also show that our method performs significantly worse when the completeness score is not optimized, which validates the effectiveness of our method.
-- In Appendix C.2, we show the performance of our method and the baseline methods when different number of concepts are retrieved. We again observe that our method significantly outperforms the baselines significantly when different number of concepts are being retrieved. We can also observe that the baseline methods (such as PCA and ACE) are not complete in both the toy example and AwA.
-- In Appendix C.3, we test how the completeness changes when a Gaussian noise with different standard deviation is added to the input in AwA, and the completeness is still high when a multivariate Gaussian noise with standard deviation 10 per dimension is added to the image (where each dimension ranges from 0 to 255).
-- In Appendix C.4, we show that the nearest neighbors of each concepts obtained on AwA when the completeness score is not optimized, which shows that some concepts are not very relevant to the animals being classified and underlining the importance of the completeness score that we introduced.
We then respond to each reviewer individually to address the questions and concerns raised.

---

### Decision · Program_Chairs · 2019-12-19

**Decision:**

Reject

**Comment:**

This paper introduces an unsupervised concept learning and explanation algorithm, as well as a concept of "completeness" for evaluating representations in an unsupervised way.

There are several valuable contributions here, and the paper improved substantially after the rebuttal.  It would not be unreasonable to accept this paper.  But after extensive post-review discussion, we decided that the completeness idea was the most valuable contribution, but that it was insufficiently investigated.

To quote R3, who I agree with: " I think the paper could be strengthened considerably with a rewrite that focuses first on a shortcoming of existing methods in finding complete solutions. I also think their explanations for why PCA is not complete are somewhat speculative and I expect that studying the completeness of activation spaces in invertible networks would lead to some relevant insights"